# Banana Pseudostem Visual Detection Method Based on Improved YOLOV7 Detection Algorithm

Liyuan Cai [1], Jingming Liang [1], Xing Xu [1,*], Jieli Duan [2,3] and Zhou Yang [2,3,4]

1   College of Electronic Engineering (College of Artificial Intelligence), South China Agricultural University, Guangzhou 510642, China
2   College of Engineering, South China Agricultural University, Guangzhou 510642, China
3   Guangdong Laboratory for Lingnan Modern Agriculture, Guangzhou 510642, China
4   School of Mechanical Engineering, Guangdong Ocean University, Zhanjiang 524088, China
*   Correspondence: xuzhexing@163.com

**Abstract:** Detecting banana pseudostems is an indispensable part of the intelligent management of banana cultivation, which can be used in settings such as counting banana pseudostems and smart fertilization. In complex environments, dense and occlusion banana pseudostems pose a significant challenge for detection. This paper proposes an improved YOLOV7 deep learning object detection algorithm, YOLOV7-FM, for detecting banana pseudostems with different growth conditions. In the loss optimization part of the YOLOV7 model, Focal loss is introduced, to optimize the problematic training for banana pseudostems that are dense and sheltered, so as to improve the recognition rate of challenging samples. In the data augmentation part of the YOLOV7 model, the Mixup data augmentation is used, to improve the model's generalization ability for banana pseudostems with similar features to complex environments. This paper compares the AP (average precision) and inference speed of the YOLOV7-FM algorithm with YOLOX, YOLOV5, YOLOV3, and Faster R-CNN algorithms. The results show that the AP and inference speed of the YOLOV7-FM algorithm is higher than those models that are compared, with an average inference time of 8.0 ms per image containing banana pseudostems and AP of 81.45%. This improved YOLOV7-FM model can achieve fast and accurate detection of banana pseudostems.

**Keywords:** deep learning; banana pseudostem; object detection; YOLOV7

## 1. Introduction

Artificial intelligence has been widely used in agriculture, with the development of agricultural mechanization and intelligence. Intelligent agricultural robots automate tedious farm work, freeing farmers from heavy work. The research hotspot of intelligent agricultural robots is the acquisition, analysis, and perception of operational information. Among them, visual information is the largest source of information for agricultural robots, which has the advantages of rich perception information and complete information collection. In the intelligent management of banana planting, the detection of banana pseudostems can be applied to the settings of counting and fertilizing banana trees. Agricultural robots with a visual perception capability benefit the intelligent management of banana plantations. Due to the dense growth of banana pseudostems in the banana plantation environment, and other pseudostem-like distractions, such as bamboo poles, it is a great challenge to implement a robust and efficient crop detection algorithm. An interesting future application is to deploy banana pseudostem detection algorithms to mobile devices, to provide the basis for applications such as the precision fertilization of banana pseudostems.

This study proposes an improved YOLOV7-based deep-learning banana false stem detection method. The main research contents are as follows.

1.   A new complex scenario-based banana pseudostem dataset was built to detect banana pseudostems;

2.  An improved YOLOV7-FM algorithm, based on YOLOV7, was proposed. Focal loss [1] was introduced, to optimize the training of complex samples, and the Mixup [2] method was used, to enhance the dataset. The two methods improved the model's generalization ability and the detection accuracy of banana pseudostems in complex environments;
3.  Based on the self-built banana pseudostem dataset, the improved algorithm YOLOV7-FM was analyzed compared to the YOLOX, YOLOV5, YOLOV3, and Faster R-CNN algorithms.

## 2. Related Work

This section reviews the development of convolutional neural networks and discusses research on banana and other plant detection.

### 2.1. Development of Convolutional Neural Networks

In the field of deep learning, convolutional neural network algorithms can be classified into three categories according to the research purpose: classification networks, detection networks, and segmentation networks. Only the development of the detection network is described here.

Detection networks can be divided into two categories, one is two-stage detector, and the other is one-stage detector. One of the two-stage detector representatives is R-CNN [3], proposed by Girshick in 2014, based on the selective search method. In 2015, He et al. proposed SPPNet [4], that is less computationally intensive and can fuse multi-scale features. Girshick proposed Fast R-CNN [5], an improved version of R-CNN and SPPNet. In 2016, He et al. proposed Faster R-CNN [6], by applying region proposal network (RPN) to Fast R-CNN. In 2017, Lin et al. proposed feature pyramid networks (FPN) [7] based on Faster R-CNN. In 2018, Liu et al. proposed an improved version of path aggregation network (PANet) [8] for FPN. Although the two-stage detection algorithm has a higher accuracy than one-stage, it runs slower and is unsuitable for mobile devices. For fast detection of banana pseudostems and mobile device deployment, the running speed of the algorithm is critical. Therefore, the two-stage detector is not used in this paper.

The one-stage detector is represented by YOLOV1 [9], proposed by Redmon et al. in 2016. This algorithm divides the image into multiple grids, and then predicts the bounding box of each grid simultaneously and gives the corresponding probability. Liu et al. proposed an SSD network containing several different detection branches, to improve the accuracy of multi-scale object detection. In 2017, Redmon et al. proposed YOLOV2 [10], and a joint training method for detection and classification. In 2018, Redmon et al. applied the idea of FPN and multiple detection branches to the network and proposed YOLOV3 [11]. In 2020, Bochkovskiy and Wang et al. proposed the backbone network of CSP-Darknet53 and introduced the PAN structure, which led to the proposal of YOLOV4 [12]. Glenn, proposed the focus structure, as well as the adaptive anchor box, based on YOLOV4, resulting in YOLOV5 [13]. In 2021, Ge et al. [14] proposed YOLOX, which decouples the regression and classification parts of the detection head and uses anchor-free. In 2022, Wang et al. [15] proposed YOLOV7, which introduced model reparametrization and dynamic label assignment, with fewer parameters and computational effort, and had faster inference and higher detection accuracy than the previous models. YOLOV7 outperforms all known object detectors in speed and accuracy in the range of 5FPS to 160FPS. Among all known real-time object detectors above 30FPS on GPU V100, YOLOV7 has the highest accuracy and is more convenient to deploy on mobile devices. Therefore, this method was used in this study

### 2.2. Research on Plant Detection

The application of computer vision and its related algorithms improves the efficiency and functionality of robots in complex agricultural environments [16–18]. Traditional detection algorithms, detect regions of interest in images based on hand-designed features and appropriate classifiers. The performance of these methods depends on the accuracy

of the extracted features and lacks robustness. With the development of deep learning technology, the application of CNN in plant disease classification, crop detection, etc., has become a hot research topic in recent years. Deep learning can extract high-level features and has better learning ability and performance than machine learning algorithms.

In terms of the one-stage detector, Gao et al. [19] integrated the CSR-DCF algorithm based on the YOLOV4 model, which improves the accurate counting of apple fruits, with mAP of 99.35%. YOLOV5 was applied to the detection and counting of peanut seedlings [20], which combined the vision transformer module on the CSPNet backbone, and the mAP of IoU = 0.5 reached 66%. YOLOX has also been applied in the detection and counting of hollyhock fruit [21]. It proposes a $\gamma$-component and Gaussian kernel convolution to improve the recognition rate of shaded and blurred samples. Sozzi et al. [22], used four versions of the YOLO model for real-time detection and counting of white grape bunches, with the best accuracy achieved by the YOLOV5X model. Wang et al. [23], developed a channel pruning YOLOV5S model, to detect apple bunches before fruit thinning, and achieved the same detection accuracy as the YOLOV5S model but with 92.7% fewer parameters. Cardellicchio et al. [24], analyzed different versions of the YOLOV5 model to identify tomatoes, flowers, and nodes, using TTA (test-time augmentation) and model ensembling methods to improve the accuracy. Dananjayan et al. [25] detected citrus leaf diseases, based on scaled YOLOV4 P7. They developed a citrus leaf dataset containing three types of diseases for disease detection, and achieved 89.3% mAP. YOLOV4 was applied to detect banana bunches and stems in [26], and banana fruits in [27]. Among them, refs. [26] achieved 99.95% AP and [27] achieved 93.69% mAP. In their banana dataset, there are fewer dense and occluded objects, and the detected objects are large and easy to identify.

In a previous study, Song et al. [28] used a Kinect depth camera to acquire 3D point cloud data of banana pseudostems, to measure their phenotypic parameters. In general, machine vision inspection methods have problems such as high prices, poor real-time performance, and severe influence by natural light. Compared with machine vision, LIDAR can overcome these problems. Xu et al. [29] used laser sensors to measure banana shoots and pseudostems. The experimental results show that the laser sensor has a strong anti-interference ability and high accuracy in the external environment under close-range measurement conditions. Jiang et al. [30] used the laser sensor to measure banana pseudostems. Then, they calculated the three-dimensional profile of banana pseudostems to obtain their phenotypic parameters. In the above studies, banana pseudostem detection was carried out under the premise that the object was artificially identified as a banana pseudostem. In contrast, the convolutional neural network can identify the object as a banana pseudostem without the premise that the object is artificially identified as a banana pseudostem. Therefore, convolutional neural networks are more advantageous for automating banana pseudostem detection. This study aims to develop a convolutional neural network-based banana pseudostem detection algorithm, to provide a technical reference for applications such as measuring the phenotypic parameters of banana pseudostems and the automatic removal of banana shoots. Previous studies on banana pseudostem detection have yet to be applied to the complex situation of dense and severe shading. The presence of a large number of occluded, dense, and small objects in a complex banana plantation environment, poses a challenge to the detection task. This study detects banana pseudostems with different growth conditions and sizes, in a complex growing environment.

## 3. Materials and Methods

### 3.1. Image Acquisition

On 15 May 2022, 687 images of banana pseudostems were taken, in complex scenes, at the Institute of Fruit Tree Research, South China Agricultural University. The image acquisition device was a color digital camera (SONY $\alpha$ 5100), with an image resolution of 6024 × 4000 pixels. The camera contains a 24.3 MP APS-C CMOS sensor. The camera exposure mode was set to auto exposure, the camera height was 160 cm, the shooting distance was about 3–10 m, and the shooting angle was horizontal. The 687 images were

divided into training and validation sets in a 7:3 ratio. The training set contained 480 images, and the validation set included 207. Due to the small amount of data on banana pseudostems in complex scenarios, data expansion was used to expand the sample of model training, to improve data diversity in subsequent studies. Validation sets were used as test sets. The experimental device used two NVIDIA RTX 3090 graphics cards. Image annotation was performed in the Labelme [31] open-source software. In the 687 images, 13,999 objects were labeled, including 9677 in the training set and 4322 in the verification set. Figure 1 shows the pictures and labels of banana pseudostems in a complex environment.

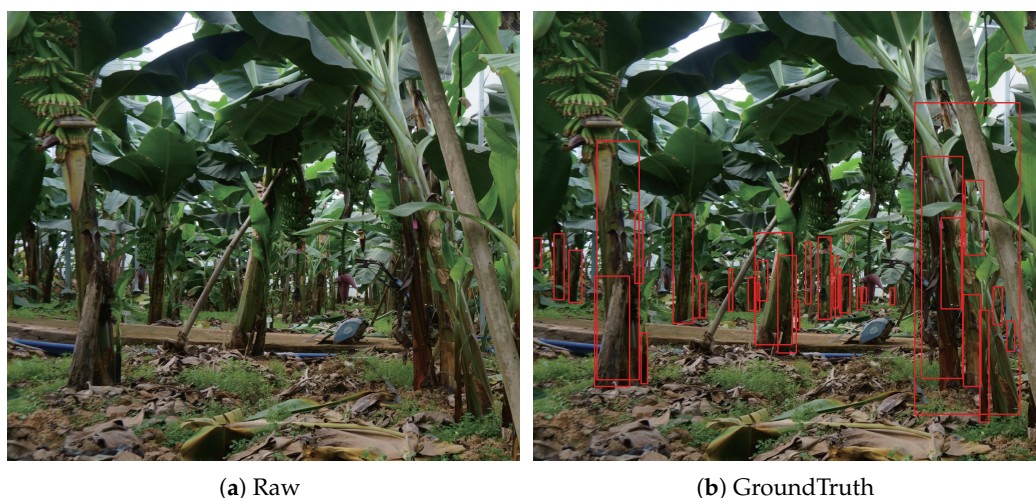

(**a**) Raw  (**b**) GroundTruth

**Figure 1.** Image of banana pseudostem cultivated area. (**a**) Original image. (**b**) The original image with pseudostems marked by red rectangles.

The labeled dataset was compared with PASCAL VOC 12 and MS-COCO object detection datasets. Table 1 gives the comparison results. From the metric "Objects/Image" in Table 1, the denseness of the banana pseudostem dataset was much higher than the PASCAL VOC 12 dataset and the MS-COCO dataset, in both the training and validation sets. Table 1 reflects the high density of the banana pseudostem dataset.

**Table 1.** Analysis with the classical object detection datasets.

| Dataset | Classes | Train | | | Validation | | |
|---|---|---|---|---|---|---|---|
| | | Images | Objects | Objects/Image | Images | Objects | Objects/Image |
| PASCAL VOC 12 | 20 | 5717 | 13,609 | 2.38 | 5823 | 13,841 | 2.37 |
| MS-COCO | 80 | 118,287 | 860,001 | 7.27 | 5000 | 36,781 | 7.35 |
| Ours | 1 | 480 | 9677 | 20.16 | 207 | 4322 | 20.88 |

### 3.2. Algorithm Description

YOLOV7 combined the reparametrization module, to replace the residual module adopted by the previous YOLO series, significantly improving the reasoning speed of the network. YOLOV7 was designed to use parameters and computational modules efficiently. The open source visualization tool PlotNeuralNet [32], can show the network structure of YOLOV7. Figure 2 shows the banana pseudostem detection neural network visualization based on the improved YOLOV7 algorithm. Complex modules in the network were simplified into a single module, for easy presentation. The improvements proposed in this paper were to the input part (Mixup) and the output part (Focal loss) in Figure 2, which were visualized as network layers. The detection process is as follows:

1. Input banana pseudostem image to the network;
2. The backbone is ELAN-Net, and features are extracted from the image using the CBS module, ELAN module, and MPC3 module;

3.  The SPPCSPC module extracts the features at different scales in the backbone;
4.  The neck is mainly composed of the FPN module [7] and PAN module [8], which merge the features of different network layers;
5.  Detection is the prediction part, which outputs the final detection results, using the features extracted from the previous layers.

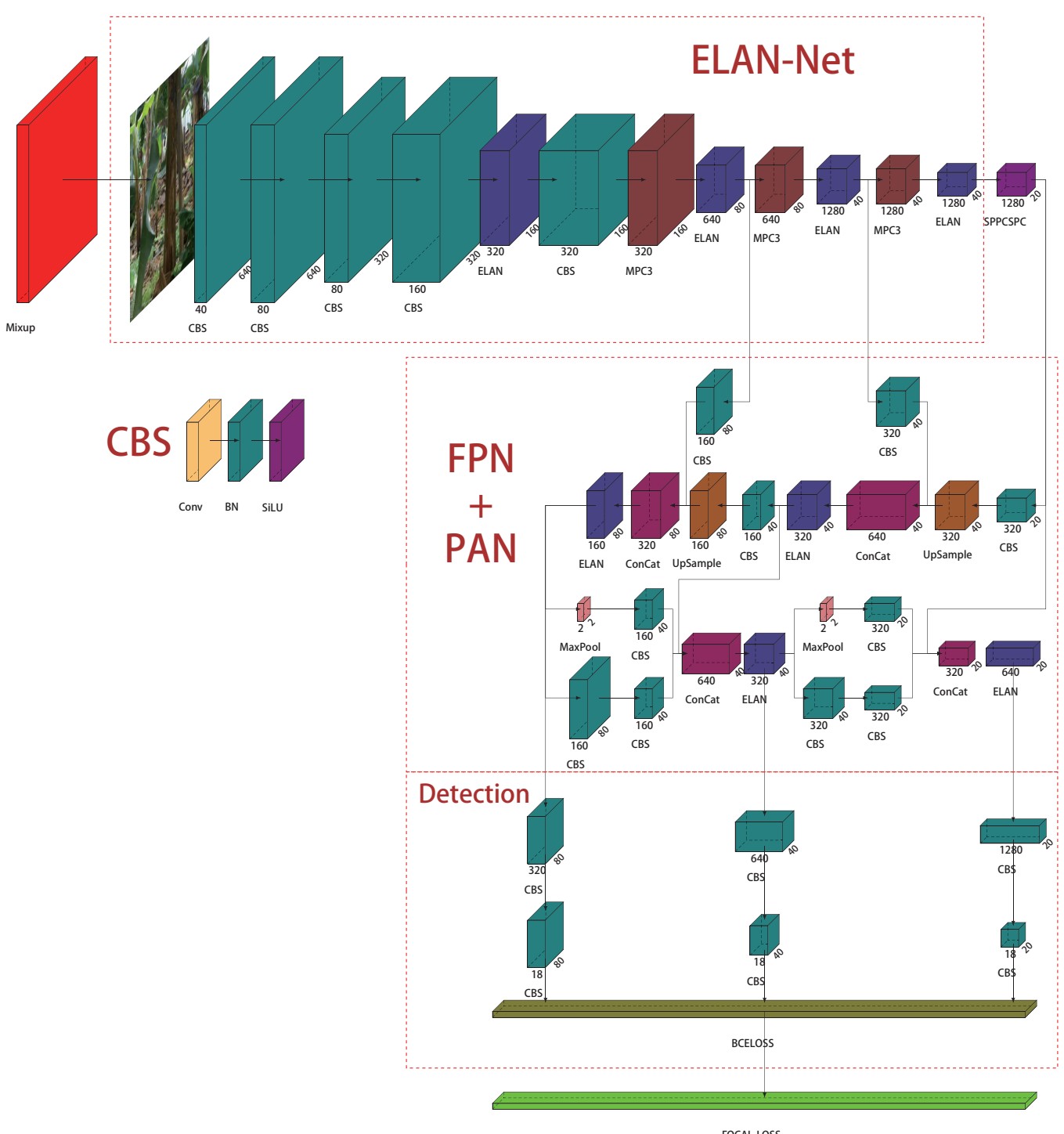

**Figure 2.** YOLOV7-FM structure.

The backbone is an ELAN-Net structure, consisting of five CBS modules (dark green blocks), four ELAN modules (dark blue blocks), and three MPC3 modules (dark brown blocks). The structure of the CBS module is composed of Convolution, BatchNormalization, and SiLU [33]. Figure 2 shows the specific structure of the CBS module. SiLU is the activation function, and the formula is as follows:

$$SiLU(x) = \frac{x}{1 + e^{-x}} \tag{1}$$

The specific structures and parameters of the ELAN module, MPC3 module, and SPPCSPC module in Figure 2, are explained in detail in Figures A1–A3, in the Appendix A. The detection part has three branches. Each branch consists of two CBS modules. These three branches detect small, medium, and large objects. YOLOV7 uses a reparametrization method. This method enhances the performance of the algorithm and tailors the network. The CBS module in YOLOV7 training uses the training method mentioned in RepConv [34].

The loss function of YOLOV7 contains three kinds of loss functions, which are classification loss, localization loss, and confidence loss. The binary cross-entropy loss function is used for the classification loss and confidence loss, which are shown in Equations (3) and (4). The CIoU loss function is used for the localization loss, shown in Equation (5). Equation (2) gives the overall loss function.

$$L(o, c, O, C, l, g) = \lambda_1 \, L_{conf}(o, c) + \lambda_2 \, L_{cls}(O, C) + \lambda_3 \, L_{loc}(l, \, g). \tag{2}$$

In Equation (2), $\lambda_1, \lambda_2, \lambda_3$ are the equilibrium coefficients of the three losses. Where $\lambda_1 = 0.7, \lambda_2 = 0.3$, and $\lambda_3 = 0.05$.

Equation (3) is the expanded form of the confidence loss.

$$L_{conf}(o, c) = -\frac{\sum_i (o_i \ln(\hat{c}_i) + (1 - o_i) \ln(1 - \hat{c}_i))}{N} \tag{3}$$

In Equation (3), $o_i \in [0, 1]$ denotes the predicted object bounding box with ground truth's IoU. $c$ is the prediction value and $\hat{c}_i$ denotes the confidence of the prediction obtained by the sigmoid function for $c_i$. $N$ is the number of positive and negative samples.

Equation (4) is the expanded form of the classification loss.

$$L_{cls}(O, C) = -\frac{\sum_{i \in pos} \sum_{j \in cls} \left(O_{ij} \ln(\hat{C})_{ij} + (1 - O_{ij}) \ln(1 - \hat{C}_{ij})\right)}{N_{pos}} \tag{4}$$

In Equation (4), $O_{ij} \in \{0, 1\}$ indicates the presence of the $j$th class of objects in the predicted object bounding box $i$. $C_{ij}$ is the predicted value. $\hat{C}_{ij}$ is the object probability obtained from $C_{ij}$ by the sigmoid function. $N_{pos}$ is the number of positive samples.

The CIoU loss is used for the localization loss. Equation (5) is the expanded form of the localization loss.

$$L_{loc}(l, \, g) = 1 - CIoU(l, g) = 1 - IoU(l, g) + \frac{\rho^2(l, g)}{c^2} + \alpha v. \tag{5}$$

$$IoU(l, g) = \frac{l \cap g}{l \cup g}. \tag{6}$$

$$v = \frac{4}{\pi^2} \left(arctan \frac{g_w}{g_h} - arctan \frac{l_w}{l_h}\right)^2. \tag{7}$$

$$\alpha = \frac{v}{(1 - IoU(l, g)) + v}. \tag{8}$$

In Equation (5), $l$ and $g$ denote the prediction box and ground truth, respectively. $\rho$ indicates the Euclidean distance. $c$ is the diagonal length of the minimum closed box

covering the two boxes. Equation (6) of *IoU* denotes the degree of overlap between the two boxes. The $\nu$ of Equation (7) means the consistency of the aspect ratio. $\alpha$ in Equation (8), is the trade-off parameter.

The officially recommended mosaic data enhancement, image mirror flip, image panning, label paste, and HSV color space enhancement, were used in the YOLOV7X training, to extend and enhance the dataset. First, the images of each epoch were randomly enhanced with HSV color. Secondly, the HSV-enhanced images were randomly panned, mirror-flipped, and scaled. Then, 15% of the labels of the images were copied to the images. Finally, four images from the current epoch were randomly taken and combined into one big picture as the input for training. Table 2 shows the super parameters of the network training. In addition, the YOLOV7X model used pre-training weights from the COCO dataset.

**Table 2.** Experimental hyperparameters.

| Hyperparameters | Value |
|---|---|
| Resolution (pixel) | $416 \times 416$, $512 \times 512$ and $640 \times 640$ |
| Batch_size | 16 |
| Steps | 9000 |
| Lr0 | 0.01 |
| Momentum | 0.937 |
| Weight_decay | 0.0005 |

The resolution in Table 2 indicates the resolution of the input image, which was generally a multiple of 32. The experimental section compares the performance of the model at different resolutions. The batch size of the training samples was 16, which was the maximum value that the device can support. Theoretically, the larger the batch size, the better. The number of steps was 9000. According to the official recommendation, the impulse and weight decay factors were 0.937 and 0.0005.

*3.3. Improvement Methods*

3.3.1. Focal Loss

Focal loss [1] was proposed by Lin et al. in 2017. It is a dynamically scaled cross-entropy loss with a dynamic scaling factor, that can dynamically reduce the weights of easily distinguishable samples during training, thus focusing the training on those hard-to-distinguish samples. In the loss calculation of YOLOV7, Focal loss was used as a processing layer (the last green layer in the YOLOV7-FM network structure in Figure 2). The formula for Focal loss is as follows:

$$FL(p_t) = -\alpha_t(1 - p_t)^\gamma log(p_t), \tag{9}$$

$$p_t = \begin{cases} p, y = 1, \\ 1 - p, other\ else. \end{cases} \tag{10}$$

$$\alpha_t = \begin{cases} \alpha, y = 1 \\ 1 - \alpha, other\ else \end{cases} \tag{11}$$

The $y = 1$ of Equations (10) and (11), denote positive samples, while the others denote negative ones. The $\alpha$ of Equation (11) is a hyperparameter used to balance the weights of positive and negative samples. The number of positive and negative samples in the detection task is highly unbalanced. The number of candidate boxes (positive samples) that can match the target in an image, is typically only a dozen or a few dozen. In contrast, the number of candidate boxes that do not match (negative samples), is much larger than that. Most negative samples not only do not contribute to training the network but also affect those samples that help training. Specifically, when y = 1, which means this sample needs to be protected, $\alpha$ should be less than 0.5, thus penalizing the class with more samples. Then,

when the sample is more unbalanced, $\alpha_t$ should be closer to 0 or 1. $p$ in Equation (10), is the prediction probability. The $\gamma$ of Equation (9), is a hyperparameter used to distinguish the classification difficulty of the samples. $\gamma$ is always greater than or equal to 0. Specifically, when y = 1, the value of $p_t$ is smaller for the hard sample, which is assumed to be 0.2. Then, the coefficient of the hard sample is $(1 - p_t)^\gamma$, which is $0.8^\gamma$. For the simple sample, the value of $p_t$ is larger, assumed to be 0.9. Then, the coefficient of the simple sample is $(1 - p_t)^\gamma$, i.e., $0.1^\gamma$. The ratio of the coefficients of the hard sample to the coefficients of the simple sample can be adjusted by setting different values of $\gamma$, to adjust the weights of the hard sample to the simple sample. This can reduce the weight of simple samples, make the loss more focused on hard samples, and prevent simple samples from dominating the whole loss function.

Applying Focal Loss to the confidence loss in the YOLOV7 loss function and to the classification loss, can solve the problems of category imbalance and differences in classification difficulty in the classification problem. The banana pseudostem dataset contains many hard samples. Focal loss can increase the weight of hard samples in the overall loss, which makes the model training focus on reducing the loss of hard samples, thus improving the recognition rate of hard samples.

### 3.3.2. Mixup

Mixup [2] is a simple and effective data enhancement method. In supervised learning, training data is used to fit real data distributions following the empirical risk minimization (ERM) principle. The authors of Mixup believed that using training data to fit the real data distribution could lead to undesirable behavior outside of the training data. Therefore, they adopted the idea of vicinal risk minimization (VRM) to construct the dataset. In short, a virtual feature target vector was generated, by sampling from a mixture of neighboring distributions, to construct new data. A set of training data is defined as $D = \{(x_i, y_i)\}_{i=1}^{n}$. After the expansion of Mixup data, a new set of data $D_v = \{(\tilde{x}_i, \tilde{y}_i)\}_{i=1}^{m}$ is obtained.

$$\tilde{x} = \lambda x_i + (1 - \lambda)x_j, \tag{12}$$

$$\tilde{y} = \lambda y_i + (1 - \lambda)y_j, \tag{13}$$

In Equations (12) and (13), $(x_i, y_i)$ and $(x_j, y_j)$ are two samples and label pairs randomly selected from the training set. $(\tilde{x}, \tilde{y})$ is the new sample constructed. Where x is the input vector of the original image. y is the one-hot label encoding. $\lambda \in (0,1)$, $\lambda$ is randomly taken from the beta distribution $\beta(\theta, \theta)$, $\theta \in (0, \infty)$. Integrating Mixup into existing training pipelines requires only a few lines of code, with little computational overhead.

Figure 3 shows the visualization of banana pseudostem data after Mixup processing. In Figure 3a,b are two images containing banana pseudostems, and (c) is a new sample generated from (a) and (b) according to Equation (12). The new sample, Figure 3c, mixes the features of Figure 3a,b, as well as the labels. In the example visualized here, hyperparameter $\lambda$, of Equations (12) and (13), is set to 0.4.

Mixup was tested on the ImageNet-2012, CIFAR-10, and CIFAR-100 datasets, to improve the generalization ability of the neural network architecture. Mixup can reduce the impact of training errors caused by corrupted labels, which can help in mislabeling and labeling bias of crops. The method can improve the robustness of the model. The mathematical principle of Mixup is straightforward. It constructs new training samples and labels by linear interpolation. Zhang et al. [2] point out that this linear modeling reduces the maladjustment in predicting data beyond the training samples. This approach helps to improve the generalization ability of the model. Mixup's improvement to YOLOV7, is to use the network's input as a processing layer (the first red layer in the YOLOV7 network structure in Figure 2).

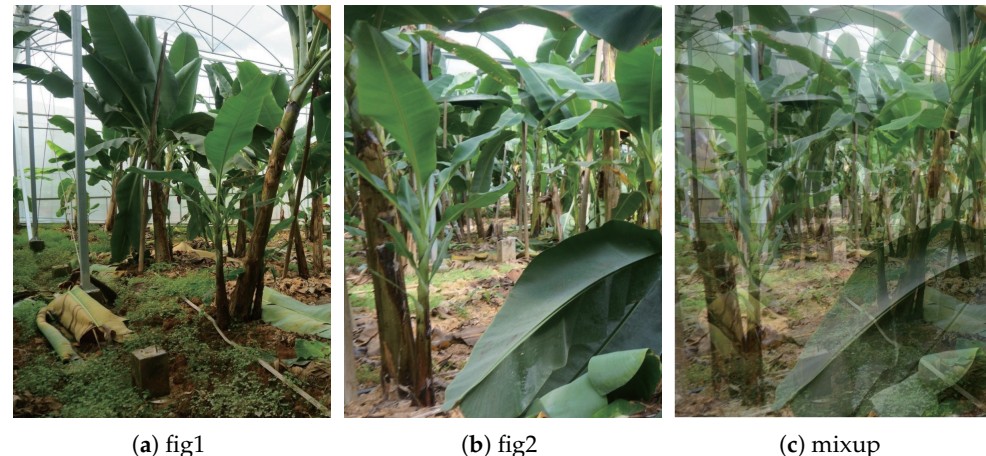

|  (**a**) fig1  |  (**b**) fig2  |  (**c**) mixup  |

**Figure 3.** (**a**,**b**) Two randomly selected samples from the training data. (**c**) The new sample after Mixup processing.

*3.4. Model Evaluation*

Precision, recall, and AP were used as evaluation metrics.

$$P = \frac{TP}{TP + FP} \times 100\%, \tag{14}$$

$$R = \frac{TP}{TP + FN} \times 100\%, \tag{15}$$

$$AP = \int_0^1 P(R)dR. \tag{16}$$

where *TP* is true positive, *FP* is false positive, *P* is precision, and *R* is recall.

After training, the training model was validated on the validation set. First, the detection objects in the validation set were iterated. Secondly, the ground truth of the detected object class was extracted, and the detection boxes of that class were read. Then, the bounding boxes below the confidence threshold were filtered out. Finally, the remaining boxes were sorted according to the confidence score, from highest to lowest. The IoU values of the box with the highest confidence level and the ground truth were analyzed. If it was greater than the IoU threshold, it was TP. Otherwise, it was FP. Missing objects were treated as false negatives. Precision and recall can be calculated to plot a PR curve by obtaining TP and FP. This study uses the PASCAL VOC metric to calculate AP, i.e., AP with IoU = 0.5. Research has considered reporting results using map 0.5:0.95. For dense and occluded datasets, there are errors in manually annotated datasets, and the map 0.5:0.95 indicator is too strict.

## 4. Results

*4.1. YOLOV7 Model Experiment*

The model version uses YOLOV7-X for training, because YOLOV7-X has a greater width and depth than the secondary version of YOLOV7, which means that YOLOV7-X can extract richer features and have better recognition effects. During the training phase, the results obtained by each epoch are validated on the validation set, resulting in a group of precision and recall rates based on thresholds. Multiple sets of precision and recall rates can be obtained when different thresholds are set for the model, allowing the P-R curve, or AP (average precision), to be plotted. There were three input scales used for training, which are 416 × 416, 512 × 512, and 640 × 640. The epoch of the training was set to 300. The best weight of AP was used as the best weight of the model. The performances of the three different input sizes were compared, where the IoU threshold was set to 0.5, and the confidence threshold was set to 0.001. Table 3 shows the final training results.

**Table 3.** Training results at different resolutions.

| Image Size | AP | P | R |
|---|---|---|---|
| 416 | 70.44% | 78.33% | 65.27% |
| 512 | 75.59% | 81.06% | 68.51% |
| 640 | 78.77% | 80.12% | 73.95% |

The AP was used to determine the comprehensive performance of the model. The precision and recall of the model with the best weights were evaluated. In Table 3, it is seen that the model's AP and recall increased as the input image resolution increased. The highest AP and recall of the model were obtained when the model input resolution was 640 × 640. All the model indexes were improved in the results of input resolution 416 × 416 and input resolution 512 × 512. When the input resolution of the model was 640 × 640, the precision decreases from 81.06% to 80.12%, compared to the input resolution of 512 × 512. The decrease in precision might be due to the increase in the model's false detection of objects, but the model's recall rate improved significantly. The accuracy results also reflect the difficulty of identifying banana pseudostems in complex scenarios in one aspect. Furthermore, the evaluation indicators for the three training processes in 300 epochs were analyzed. Figure 4 shows the changes in the metrics during the training.

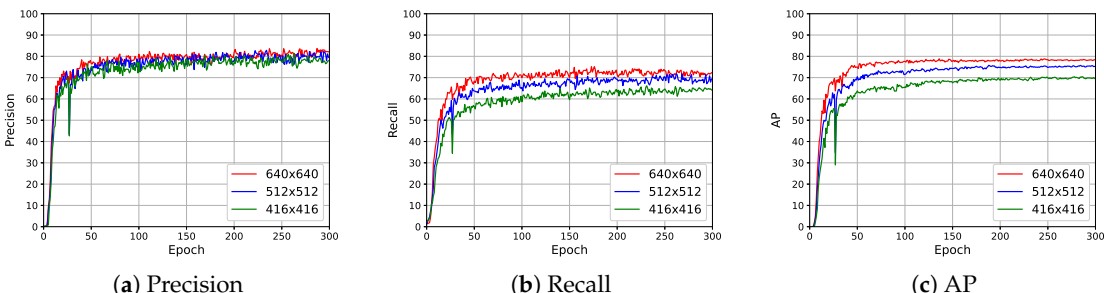

(**a**) Precision      (**b**) Recall      (**c**) AP

**Figure 4.** Iteration process of metrics with different resolutions. Figures (**a**–**c**) show the iterations of precision, recall, and AP metrics during training, respectively.

As can be seen in Figure 4, the model fit quickly in the early epochs of training, due to the use of loaded pre-training weights. The model metrics then increased slowly from the 50th epoch to the 100th epoch, and are relatively stable from the 100th epoch to the 300th epoch. The AP performance of the model was better for the input resolution of 640 × 640 compared to the input resolutions of 512 × 512 and 416 × 416. Therefore, in the subsequent comparison experiments, the 640 × 640 model input was used as the basis for model improvement and comparison experiments.

The banana pseudostem dataset contains a very dense setting, and includes small and occluded objects. These factors posed a significant challenge to the performance of the model. An example is given in Figure 1b, to demonstrate these situations.

A scale differential experimental analysis was performed on the model, to analyze the detection effect of banana pseudostems in more detail. Specifically, the object area in the test set was divided into small, medium, and large-scale objects. Small objects, were those smaller than 32 square pixels. Medium objects, referred to objects larger than 32 square pixels and smaller than 96 square pixels. Large objects, referred to objects larger than 96 square pixels. Table 3 gives the results of the scale differentiation of YOLOV7 (input resolution of 640 × 640).

In Table 4, the AP of the model was low for small and medium objects. The AP of small object detection was only 32.5%. The objects of other scales easily sheltered small and medium objects, which contained fewer pixels. The features available were fewer compared to the objects of other scales. Among them, small object detection was one of the difficulties in deep learning object detection and was the main reason for the low accuracy of the object detection model. Two ideas were proposed to solve the problem of low accuracy for small and medium-sized objects in banana pseudostem detection.

**Table 4.** Accuracy of objects at different scales.

| Object Scale | AP |
|---|---|
| Small | 32.5% |
| Medium | 68.9% |
| Large | 91.1% |

1.  Using Focal loss to improve the loss function allows the model to focus on training hard samples, as training for the occlusion of small- and medium-sized objects was challenging;
2.  In the complex orchard environment, some specific banana pseudostems could not be identified, or were identified with low accuracy. Mixup data enhancement could be used to try to improve the generalization ability of the model.

*4.2. Improved Experiments*

**Focal loss.** The authors of [1] experimented with the effect of multiple sets of $\gamma$ and $\alpha$ on the COCO dataset. The results showed that AP performed best at $\gamma = 2$, $\alpha = 0.25$. Without fine-tuning it, this study simply used hyperparameters $\gamma = 2$, $\alpha = 0.25$. The $\gamma$ of Equation (12) was set to 2.0, and the $\alpha$ of Equation (14) was set to 0.25, to balance the loss. Figure 5 shows the AP iteration curve of the model after the Focal loss improvement. Due to pre-training weights, the improved model fit quickly in the epochs in the early training period. The AP curve of the model improved by Focal loss was lower than the baseline in the first 100 epochs of training. The reason was that Focal loss increases the weight of hard samples in the loss, and the model slows down the fit of standard samples. In the later stage of model training, from the 150th epoch to the end of the training, the AP curve of the model improved by Focal loss gradually exceeded the baseline.

Focal loss improved the model. No. 5 in Figure 6b was a pseudostem heavily obscured by plantain leaves. No. 6 was a pseudostem that was more similar to lichen. Both no. 5 and no. 6 were hard samples. Pseudostems no. 5 and no. 6 were not detected in baseline, but they both could be detected after the Focal loss improvement. However, pseudostems no. 1 and no. 7, which were not detected in baseline, were also not identified after the Focal loss improvement. No. 1 was a partially obscured pseudostem, and the image features resembled the bamboo in the pseudostem environment. No. 7 was a small pseudostem, which was more heavily obscured and could provide fewer features. No. 4 was also a small pseudostem and was not labeled in the dataset, which could be interpreted as an unlabeled sample. The baseline model did not recognize this no. 4 pseudostem. After the improvement of Focal loss, the no. 4 pseudostem could be identified, even though it was unlabeled. Pseudostems no. 1 and no. 7 were still not detected. Although the Focal loss slightly reduced the confidence in identifying some normal samples, it improved the ability of the model to identify hard samples.

**Mixup.** The Mixup paper tested a set of alpha experiments on CIFAR-10 data, when 20% of the data were replaced by random noise. The conclusion showed that the test error was lowest when $\theta = 8$. The $\theta$ parameter could be fine-tuned, but this study did not do so. The hyperparameter $\theta = 8$ was used in the experiment. In the Mixup experiment, the $\lambda$ in Equations (15) and (16) are randomly taken as beta distribution $\beta(8,8)$. Mixup data enhancement was performed on 40% of the training samples for each epoch, when training the network. The data enhancement strategy was to start during the training, not before the training. Therefore, the above data enhancement was not through increasing the number of training datasets but through dynamic data enhancement during training. Figure 7 shows the AP iteration curves of the model after the Mixup data enhancement. The model was fitted quickly in the first dozen epochs early in training. The AP curves of baseline and Mixup in the first 50 epochs were very similar. The Mixup curve gradually exceeded baseline from the 50th epoch.

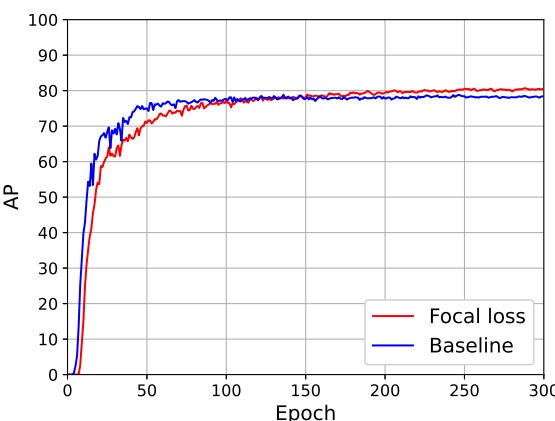

**Figure 5.** Focal loss AP.

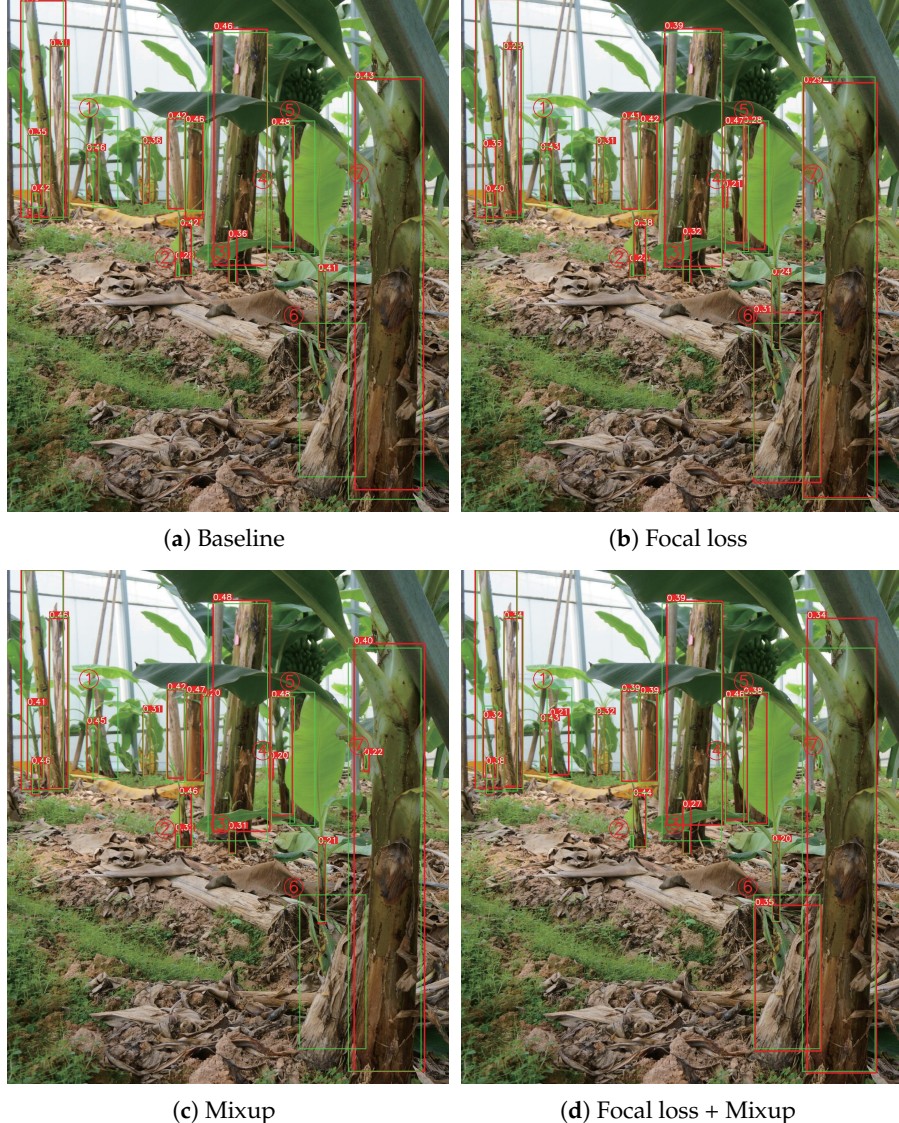

(**a**) Baseline                                                                                                    (**b**) Focal loss

(**c**) Mixup                                                                                                       (**d**) Focal loss + Mixup

**Figure 6.** An example of improved image recognition. Here, the red boxes indicates the detection boxes. The green boxes represent the ground truth. (**a**–**d**) The seven pseudostem positions marked with red round numbers. The marked positions are in the upper left corner of the pseudostem positions. All the numbers have their corresponding ground truth except number 4, which has no ground truth.

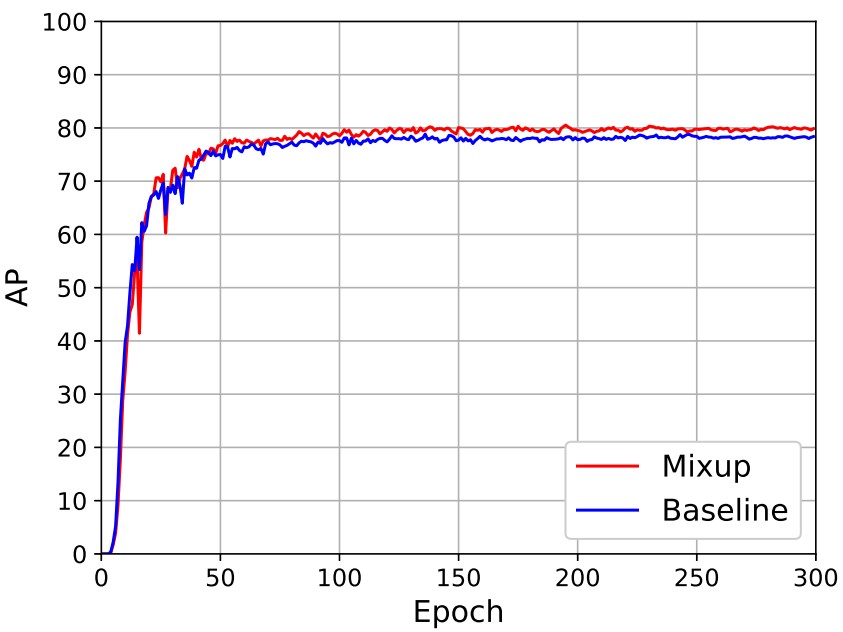

**Figure 7.** Mixup AP.

Mixup improved the robustness and generalization ability of the model. The detection box regression of the small pseudostem of no. 3 in Figure 6c matched better with baseline. After the improvement of Mixup, the model was able to identify pseudostem no. 4 and pseudostem no. 7. However, the pseudostems of no. 1, no. 5, and no. 6 were still not identified. These two improvement ideas (i.e., Focal loss and Mixup) were combined, to improve the recognition ability of the model.

**Comprehensive improvement.** A mixed experiment of Focal loss and Mixup was attempted in this experiment. Compared with the Focal Loss and Mixup experiments, the mixed experiment achieved the best results. Figure 8 shows the AP iteration curve of the model training. In Figure 8, the best curve (red curve) fit at a similar speed to the one improved by Focal loss only (blue curve). The best curve (red curve) slowed the fit to normal samples early in training, as the model focused on training hard samples. The combined improved model fit slower than the one fit by Mixup and baseline only. The AP curves of the best model gradually fit in the first 150 epochs and level off and increased slightly from the 150th epoch to the end of training. The AP of the best model was higher than Focal loss, Mixup, and baseline.

The best model achieved results beyond Focal loss and Mixup. Figure 6d shows that the best model could detect the previously undetected pseudostem no. 1. Although the regression of pseudostem no. 1 was not a good match, it was much better than not being detected. The regression of the detection box of pseudostem no. 3 was better than that of Mixup in Figure 6c. The banana leaves of other pseudostems obscured the upper part of pseudostem no. 3. Because of the obscuration, it was debatable whether the top part, above pseudostem no. 3, belonged to no. 3. Therefore, the return of the test box of pseudostem no. 3 was not important here. The best model can detect pseudostem no. 5, which was heavily obscured. Pseudostem no. 6, similar to lichens, could also be detected. Unfortunately, small pseudostems no. 2 and no. 7 were not detected. The detection of small pseudostems was still a great challenge for the model.

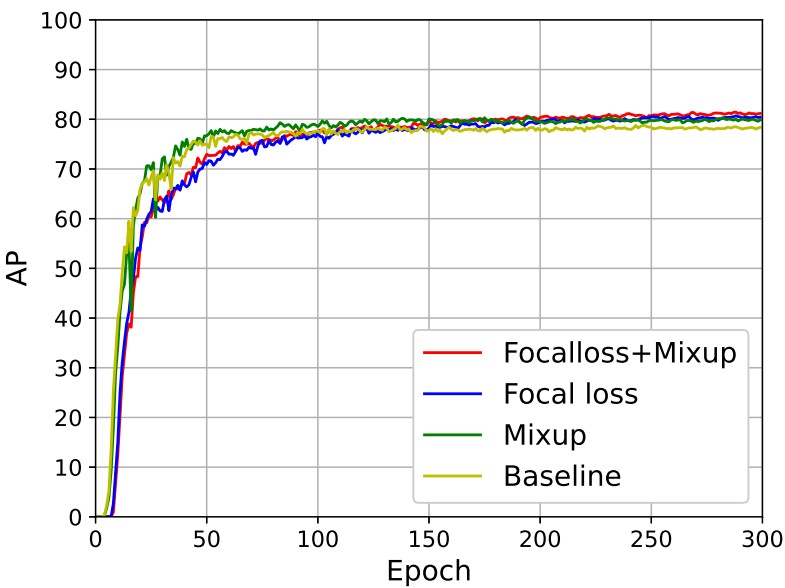

**Figure 8.** Comprehensive experiment AP.

To better analyze the detection effect of the pseudostems, this work analyzed the object scales of detection for the models with the Focal loss improvement, Mixup improvement, and combined improvement. This work evaluated the detection at three scales: large, medium, and small in the dataset. Table 5 shows the results of the experiment.

**Table 5.** AP of banana pseudostems with different scales.

| Object Scale | Baseline | Focal Loss | Mixup | Best |
|---|---|---|---|---|
| Small | 32.5% | 37.8% | 37.2% | 38.7% |
| Medium | 68.9% | 72.9% | 70.9% | 73.9% |
| Large | 91.1% | 92.7% | 91.7% | 93.4% |
| Average | 78.77% | 80.56% | 80.53% | 81.45% |

All experiments showed improved AP values for object detection at all three scales. Among all the improved experiments, small objects had the largest AP gain, followed by medium objects, and finally large objects. Among them, the best model had the largest AP gain. The best model, improved by Focal loss and Mixup, achieved 81.45% AP, among which large object AP achieved 93.4%, medium object AP reached 73.9%, and small object AP achieved 38.7%.The AP of the three scales of the best model exceeded the AP of the baseline, and the models improved by Focal loss only and Mixup only. In Table 5, both Focal loss and Mixup improved the detection accuracy for different scales of objects, especially for small and medium objects, proving the experiment's validity.

## 5. Discussion

Although the improved experiment was practical, it had yet to compare to other models, and the experimental effects of data in other environments had yet to be tested. This section provides additional experiments to improve the integrity of the experiment. Table 6 shows the experimental results of the YOLOV3, YOLOV5, YOLOX, YOLOV7-X, and Fast R-CNN models. In the comparison experiment, YOLOV5 used the YOLOV5-X-V6.1 version. YOLOX used the YOLOX-X version. The backbone of Faster R-CNN was ResNet50, and the neck part was the FPN structure. YOLOV7 adopted the YOLOV7-X version. The input resolution of the YOLO series model was set to 640 × 640 by default. The input resolution of Faster R-CNN was variable, and the variation interval was (800, 1333; where

800 represents the minimum edge and 1333 represents the maximum edge). As shown in Table 6, the AP of YOLOV7-FM reached the highest, 81.45%.

**Table 6.** Training results of different models.

| Model | AP |
|---|---|
| Faster R-CNN | 64.8% |
| YOLOV3 | 71.46% |
| YOLOV5-X | 76.18% |
| YOLOX-X | 72.8% |
| YOLOV7-X | 78.77% |
| YOLOV7-FM | 81.45% |

To represent the performance of banana pseudostems at different scales in different models, this work selected the YOLOV5-X model, with the next best AP performance, to compare with YOLOV7-X and YOLOV7-FM. Table 7 shows the results of the comparison. From the results in Table 7, it can be seen that, although YOLOV5-X achieves 90.1% AP performance in large-scale banana pseudostems, the AP performance in small and medium-scale banana pseudostems is much lower than YOLOV7-X and YOLOV7-FM.

**Table 7.** AP of YOLOV5-X, YOLOV7-X, and YOLOV7-FM at different banana pseudostem scales.

| Object Scale | YOLOV5-X | YOLOV7-X | YOLOV7-FM |
|---|---|---|---|
| Small | 29% | 32.5% | 38.7% |
| Medium | 63.1% | 68.9% | 73.9% |
| Large | 90.1% | 91.1% | 93.4% |

To clarify the gap between the models, this work compared the number of parameters and FLOPS (floating-point operations per second) and inference time of the above models, based on RTX 3090 as the inference device. Table 8 shows the specific parameters.

**Table 8.** Comparison of model parametric numbers, GFLOPS (giga floating-point operations per second), and inference time per image.

| Model | Parameters | GFLOPS | Inference Time (ms) |
|---|---|---|---|
| Faster R-CNN | 4135000 | 206.7 | 35.7 |
| YOLOV3 | 61497430 | 154.7 | 36.4 |
| YOLOV5-X | 86173414 | 204 | 12.5 |
| YOLOX-X | 99000000 | 118.9 | 23.9 |
| YOLOV7-X | 70815092 | 188.4 | 8.0 |
| YOLOV7-FM | 70815092 | 188.4 | 8.0 |

The inference times in Table 8 included model output and NMS (non-maximum suppression) time. Although Faster R-CNN had the smallest number of parameters, it had more GFLOPS than all models in the YOLO series, and inference was, surprisingly, slightly faster than YOLOV3 (we guess that this was related to the density of the test data). Among the YOLO series models, YOLOV7-FM had the fastest inference time, with fewer parameters than YOLOX and YOLOV5, and was slightly higher than YOLOV3. YOLOV7 differed from the other models, in that YOLOV7 used a heavily parametrized structure, eliminating many branching structures in the model (e.g., ResNet connections). YOLOV7 achieved the fastest inference time, proving the effectiveness of applying the heavy parametrization structure in the YOLOV7 model. Compared with YOLOV7-X, YOLOV7-FM achieved higher accuracy, without enhancing the number of parameters, GFLOPS, and inference time of the model.

To demonstrate the detection performance of the improved model in different banana pseudostem scenarios, this study explored the model's effectiveness in detecting banana pseudostems that are about to be recycled as fertilizer. Figure 9 shows the detection results for banana pseudostems with their crowns cut off. The banana pseudostem in Figure 9a could be identified. In Figure 9b, banana pseudostems in the near distance could

be recognized, but not those that were slightly further away. Figure 9 shows that the YOLOV7-FM model has good generalization ability in other scenes.

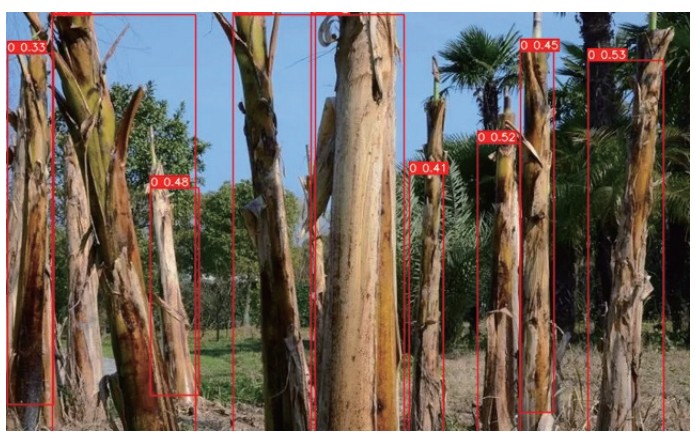 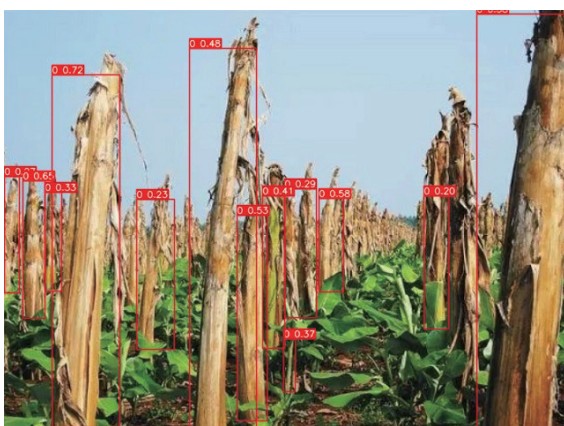

(**a**) Scenario 1                                         (**b**) Scenario 2

**Figure 9.** Effectiveness of detecting banana pseudostems in other test scenarios. (**a**,**b**) Banana pseudostems to be recycled as waste, with the canopy trimmed off.

Although the improvements were effective, there might be better ways to make them more effective. In recent years, research on oriented object detection has received widespread attention. Oriented object detection might be better than rectangular box detection. Compared to rectangular boxes, oriented boxes could take into account the object's angle and cover the object better. However, this would increase the cost of labeling, because oriented and rectangular boxes are two different labels. In a dense banana pseudostem detection setting, oriented object detection might be a better idea.

## 6. Conclusions

The fast and accurate detection of banana pseudostems is important for the intelligent management of banana planting. This paper proposed an improved YOLOV7-based method for detecting banana pseudostems in complex orchard environments. This study analyzed the detection performance of banana pseudostems at different resolutions and the problems of shelter and different object scales in the banana pseudostem dataset. Two improvements proposed in this study were Focal loss and Mixup. Based on the experimental results, the following conclusions can be summarized:

1. This study found a suitable deep-learning algorithm for banana pseudostem detection and analyzed the structural features of the YOLOV7-X network and some problems in banana pseudostem detection;
2. This study improved YOLOV7-X by using Focal loss and Mixup data enhancement. The improved YOLOV7-FM model could better identify small- and medium-sized banana pseudostems. YOLOV7-FM had better recognition of banana pseudostems that were heavily obscured and banana pseudostems that were highly similar to the environment;
3. This study compared YOLOV7-FM with YOLOV3, YOLOV5, YOLOX, YOLOV7-X, and Faster R-CNN, and analyzed the number of parameters, GFLOPS, and inference times of the different models. YOLOV7-FM achieved the highest accuracy and inference speed.

The results show that the method is suitable for detecting banana pseudostems in complex orchard environments. There are some areas for improvement in this work, such as, the model found it difficult to discriminate banana pseudostems that were entirely obscured by banana leaves. Such completely obscured banana pseudostems do exist in the natural environment. However, the small number of them, and the need for more sufficient data for model training, made this case difficult. Future work will deploy the

model on mobile devices, such as NVIDIA JETSON NANO, and use a depth camera to obtain the pixel distance of the banana pseudostems, which can provide accurate distance information, for the fertilization device to adjust the relevant parameters.

**Author Contributions:** Conceptualization, L.C.; methodology, L.C.; software, L.C.; validation, L.C. and J.L.; formal analysis, L.C. and J.L.; investigation, L.C.; resources, X.X. and J.D., and Z.Y.; data curation, L.C. and J.L.; writing—original draft preparation, L.C.; writing—review and editing, L.C. and X.X. and J.D., and Z.Y.; visualization, J.L.; supervision, X.X.; project administration, X.X.; funding acquisition, X.X. and J.D., and Z.Y. All authors have read and agreed to the published version of the manuscript.

**Funding:** This research was funded by the Laboratory of Lingnan Modern Agriculture Project, grant number NT2021009, and the China Agriculture Research System of MOF and MARA, grant number CARS-31.

**Data Availability Statement:** The data presented in this study are available on request from the corresponding author. The data are not publicly available due to the multiple interests involved in the data.

**Conflicts of Interest:** The authors declare no conflict of interest. The funders had no role in the design of the study, in the collection, analyses, or interpretation of data;,in the writing of the manuscript, or in the decision to publish the results.

## Abbreviations

The following abbreviations are used in this manuscript:

| | |
|---|---|
| AP | Average precision |
| CNN | Convolutional neural network |
| IoU | Intersection over union |

## Appendix A

Figure A1 shows the specific structure of the ELAN module. The ELAN module is an innovative and efficient feature aggregation module of YOLOV7, with enhanced robustness. The input first passes through the first branch of the CBS module, containing six CBS modules with stride one and convolution kernel size of $3 \times 3$, to extract features. The input then passes through a second branch, containing a CBS module with a convolution kernel of size $1 \times 1$, to scale the channel for subsequent splicing operations. Finally, the output features of layers 1, 3, and 5 of the first branch, are concatenated with the output features of the two branches, to obtain the final feature extraction result. This modular design enables the ELAN module to enhance the learning capability of the network without destroying the original gradient path.

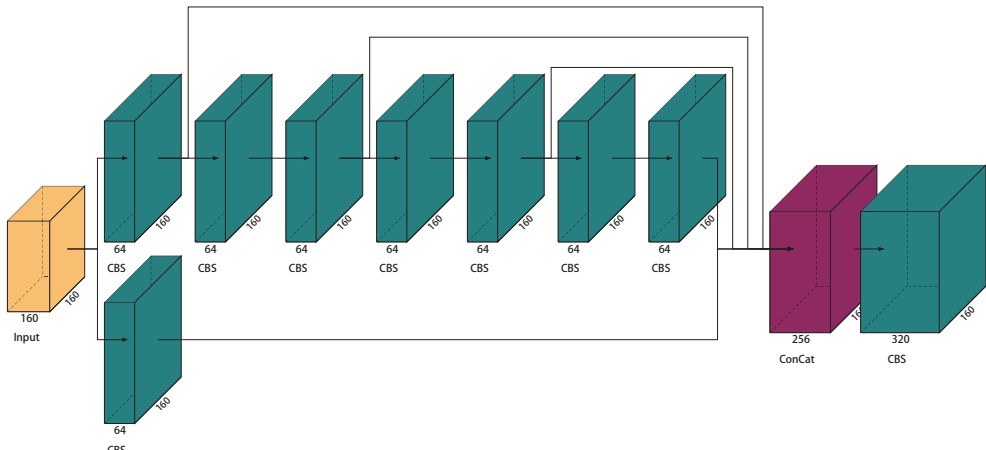

**Figure A1.** ELAN Module.

Figure A2 shows the specific structure of the MPC3 module. The MPC3 module consists of one MaxPooling and three CBS modules. The input first passes through the first branch, containing one MaxPooling layer of $2 \times 2$, and a CBS module with a stride size of 2 and a convolutional kernel size of $3 \times 3$. The input then passes through a second branch, which contains a CBS module with a convolutional kernel size of $1 \times 1$ and a CBS module with a step size of 2 and $2 \times 2$ convolutional kernels. Finally, the feature layers from the two branches are concatenated together, to obtain the downsampled features.

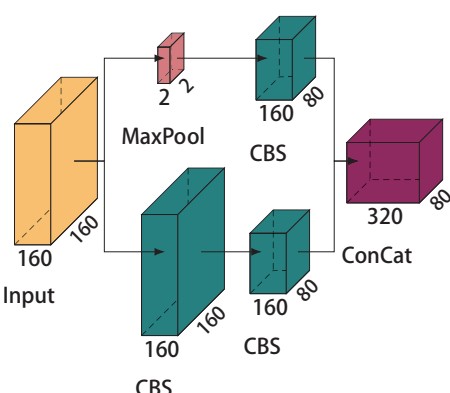

**Figure A2.** MPC3 Module.

Figure A3 shows the specific structure of the SPPCSPC module. The SPPCSPC module has two main branches. In the first branch, the input passes through three CBS modules, whose convolution kernel parameters are $1 \times 1$, $3 \times 3$, and $1 \times 1$, and their strides are all 1. Finally, the output features of the four MaxPooling layers are concatenated and output to a CBS module of size $3 \times 3$, and then passed through a CBS module of $1 \times 1$, where the stride size of both CBS modules is 1. The second branch is a $1 \times 1$ CBS module. Finally, the output features of the two branches are concatenated together, to obtain the features with different receptive fields.

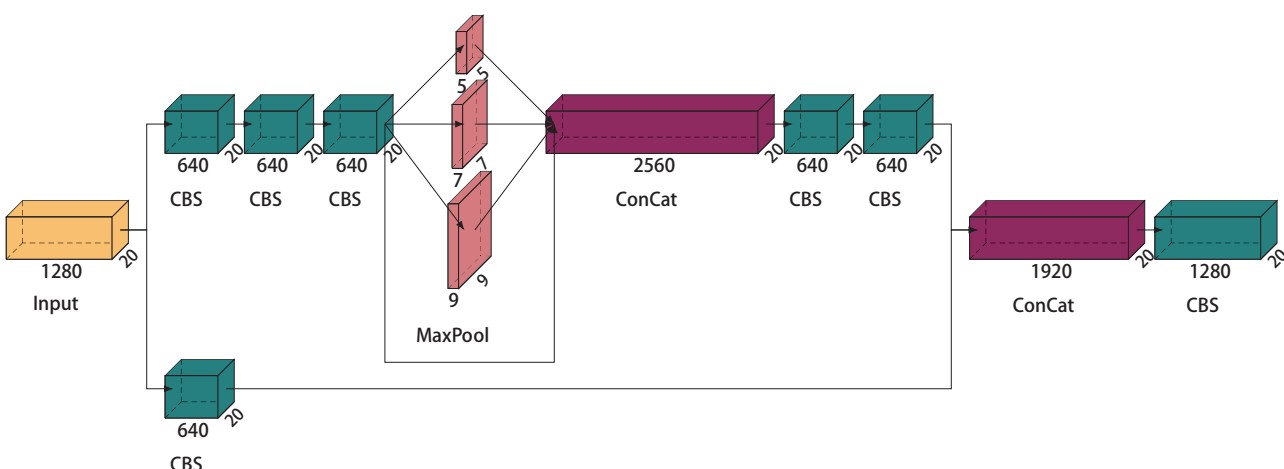

**Figure A3.** SPPCSPC Module.

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
