# Peer review of "Banana Pseudostem Visual Detection Method Based on Improved YOLOV7 Detection Algorithm"

_agronomy, doi:10.3390/agronomy13040999_

Round 1

Reviewer 1 Report

Please find attached the review.

Author Response

Please see the appendix.

Reviewer 2 Report

The authors present in the manuscript an interesting application and correct analysis of the YOLOV7 detection algorithm to identify banana pseudostem by image analysis in complex cops  environments.

The paper needs to be improved, therefore I recommend the following suggestions

Title:

I recommend: Banana pseudostem visual detection method based on improved

YOLOV7 detection algorithm

Abstract:

Line 4 and 10: “the complex environments”. Describe the characteristics of a complex crop or environment. Why is it considered a complex environment?

Line 8: “hard simples”. why are they considered hard samples?

Introduction:

Line 30: The idea “In the future, we intend to deploy…”, change for: An interesting future application is to deploy……”

Line 33 to 47: This should be moved to the methods and materials section.

In each item, rewrite the text expressed in the third person. For example: “1- A new complex scenario-based banana pseudostem dataset for banana pseudostem detection was built”

Note: Revise the entire manuscript, removing the sentences "we" "our"

Line 122: “To the best of our knowledge, there is no research on using convolutional neural networks for banana pseu-dostem detection.” This is irrelevant, delete the sentence

Line 136: with a resolution of 6024×4000, Replace by “image resolution 6024×4000 pixel”

Figure 1: I recommend reviewing the title, by “Image of the cultivated area… or area of interest

Line 157: “50% of the computational effort of the model” What computational effort or time does it refer to?

The description and analysis of results is coherent and adequate. Maybe I should improve the discussion

Line 422: this should refer to table 7, review the numbering of the following tables and check the text of the manuscript.

Note: In tables 4, 5, 6 and 7 delete the average row (or item), confuse the reading

The grammatical aspect of the manuscript must be seriously reviewed

Author Response

Please see the appendix.
